# Building a Patient-Centered Weight Management Program: A Mixed Methods Project to Obtain Patients’ Information Needs and Ideas for Program Structure

**DOI:** 10.3390/pharmacy7040165

**Published:** 2019-12-03

**Authors:** Austin Arnold, Erin Holmes, Meagen Rosenthal

**Affiliations:** Department of Pharmacy Administration, School of Pharmacy, University of Mississippi, P.O. Box 1848, Oxford, MS 38677, USA; adarnol1@go.olemiss.edu (A.A.); erholmes@olemiss.edu (E.H.)

**Keywords:** weight management, obesity, community-based participatory research, patient-centered programming, community pharmacy

## Abstract

Achieving and maintaining weight loss for large segments of the population remains elusive, despite evidence demonstrating the value of many weight management programs. This study aimed to gather patients’ perceptions on weight management education needs, and ideas for the structure of a weight management program to be delivered in community pharmacies. This was an exploratory mixed methods study combining qualitative focus group interviews with a cross-sectional survey. Three focus group interviews were conducted, along with a brief survey based on focus groups findings and sent to all eligible participants. The survey allowed for individual responses on the program components and narrowing down of focus group findings. Nearly half of the respondents (45.9%) wanted further education on limiting carbohydrate and sugar intake. Participants were most interested in identifying different exercises appropriate for those with physical limitations (48.6%). Most participants preferred 1 h meetings (70.3%) that contain a mix of one-on-one and group meeting formats (67.6%). The results of the study suggest a three-month weight management program, with a combination of group and individual in-person meetings occurring twice per month, would be of most interest to patients.

## 1. Introduction

Obesity is a leading health concern around the world. A recently published study of body mass index (BMI) trends illustrates that for the first time in history there are now more people in the world who are overweight than underweight [1]. In fact, approximately 13% of the world’s adult population has been diagnosed with obesity [2], and more than 30% of the United States’ (US) adult population face the same diagnosis [3]. Furthermore, obesity is a major risk factor for chronic diseases such as heart disease, diabetes, high blood pressure, and cancer [2]. Consequently, obesity contributes to three of the top five highest reported causes of death in the US [4].

According to the September 2015 Robert Wood Johnson Foundation Trust for American’s Health report, rates of obesity now exceed 35% in five states: Mississippi, Louisiana, Alabama, Arkansas, and West Virginia [5]. Obesity and its complications also disproportionately impact minority populations with 48.1% of all African American adults and 42.5% of all Hispanic adults being diagnosed with obesity [6]. Given that these groups represent a significant proportion of the overall US population, developing targeted strategies to address the obesity epidemic is imperative [7].

Fortunately, there have been many studies that have led to the development of successful weight management programs, which can be implemented in a variety of practice settings [8]. One example of such a program is the Diabetes Prevention Program (DPP), wherein patients participated in a 16-session intensive lifestyle intervention program with goal setting, nutrition and exercise counseling, and behavioral trouble-shooting [9]. Patients in this program lost a mean of 7 kg in the first year of the program, and were able to maintain their weight below baseline measures at the 10 year follow-up of the study [10]. Despite the volume of evidence for the negative outcomes of obesity and evidence of successful weight management programs, weight loss and weight loss maintenance for large segments of the population remain elusive. Every year approximately 45 million American adults attempt to lose weight [11]. A systematic review found that only 54% of patients who initially lost weight were able to maintain that loss [12]. Evidence also demonstrates that the more dramatic the initial weight loss, the more dramatic the subsequent weight re-gain [13], and this weight re-gain can begin within as little as 6 months of the initial weight loss [14].

Compounding this problem is a lack of access to the resources and information patients need to tackle and manage this condition. Research shows a significant decline since 1996 in the chances of patients receiving weight or diet/exercise counselling from physicians [15]. Importantly, this decrease was most pronounced in patients diagnosed with conditions such as hypertension, diabetes, and obesity [15]. In short, individuals with chronic diseases who would benefit most from weight management counselling are not receiving it from their physician. Moreover, these access issues are worse in states like Mississippi where all but two of its 82 counties are designated rural [16], and the majority of the state is considered medically underserved [17].

Pharmacists are widely regarded by patients as being among the most trustworthy and accessible health care professionals [18]. A recent report from North Carolina revealed that high risk Medicaid patients visited community pharmacies 35 times in 2013, compared to only 3.5 visits to other primary care providers [19]. As such, pharmacists have more opportunities to engage with patients, especially in rural areas where other health services are often limited [20]. Patients have also shown willingness, and interest, in receiving weight management services from pharmacists in the community setting [21,22].

Moreover, the Accreditation Council for Pharmacy Education (ACPE) Standards requires pharmacists be trained to interpret laboratory results, perform health screenings, and provide basic physical assessments [23]. Pharmacists also learn how to design and implement national or community-based public health, wellness, and prevention programs [23]. This baseline knowledge can be easily paired with specific training on basic dietary counselling, physical activity guidance, and behavioral counseling prior to providing a weight management program [23]. Previous research evaluating such community pharmacy-led weight management programs demonstrated pharmacists’ ability to assist patients with a mean weight loss of 3.8 kg across several studies [24,25,26,27,28,29,30,31,32].

Along with access issues, previous research into patients’ perspectives of weight loss identified both barriers and facilitators. Some barriers include factors such as stress, depression, and food cravings, external barriers such as lack of time, and internal barriers such as lack of motivation [33]. A qualitative analysis of six focus group interviews with African American women revealed additional barriers including lack of access to resources, issues with self- and extrinsic control, and identification with a larger body size [34]. Other barriers included life transitions, health status change, lack of accountability, and an absence of social support [35]. Some facilitators from the patients’ point of view included accountability to others, social support, planning ahead, awareness and mindfulness of food choices, basic nutrition education, portion control, exercise, and self-motivation [35]. Other facilitators from a study focusing on African American adults included creating specific goals and strategies, offering a program with access to healthy foods, expanding community resources, and streamlining healthcare services [36].

While barriers and facilitators are obviously important to patients, they have yet to be actively integrated into a weight loss/management program and subsequently tested for efficacy with other programs. Furthermore, patients have not been queried about the structure of a weight management program for delivery in a community pharmacy setting. According to the Patient Centered Outcomes Research Institute (PCORI), research that is driven by the interests and needs of patients is more likely to be taken up and used by those patients in addressing individual health needs [37]. Therefore, by accounting for patients’ needs and wants in the design of a weight management program, patients will ultimately be more likely to engage with the program and successfully manage their weight. As such, the objective of this study was to gather patients’ weight management education needs and ideas for the structure of a weight management program to be developed and tested in a community pharmacy setting. The specific aims of the study were: (1) identify all potential aspects of a weight management program that meet the needs of patients; (2) determine the most important aspects for a weight management program; (3) propose potential weight management programs that meet the stated needs of patients.

## 2. Materials and Methods

### 2.1. Study Design

This was an exploratory mixed methods study combining qualitative focus group interviews with a cross-sectional quantitative survey each designed to meet the study objective. An exploratory mixed methods design was appropriate to meet the study objective because prior research did not assess patients’ needs when developing a weight management program.

### 2.2. Participants and Recruitment

Focus group interview participants were at least 18 years of age, self-reported to be overweight or have obesity, and expressed interest in a weight management program delivered in community pharmacies. While the researchers acknowledge that weight status perception varies between cultural groups, self-reported interest in participating in a community pharmacy-based weight management program is more important for this phase of the research where advice and structural design ideas are being collected [34].

From December 2018 to January 2019, English-speaking patient participants, from an existing research collaborative focused on diabetes self-management, were recruited. Participants from this collaborative were invited to attend a focus group discussion in their home community and encouraged to bring a friend who was also interested in weight management. The focus group interview sessions included a discussion on the ideal weight management program, refreshments, and a $25 gift card as compensation for participation.

Participants for the cross-sectional survey were recruited from the same research collaborative as the focus group interview participants, as well as any additional participants brought to the focus group interviews by previous collaborative members. Participants who returned completed surveys received a $5 gift card as compensation for participation. The University of Mississippi Institutional Review Board approved all procedures prior to participant recruitment and initiation of the focus groups and survey (IRB #: 18x-129).

### 2.3. Focus Group Data Collection Procedures

The focus group interviews were conducted in each of the three communities participating in the research collaborative. Each focus group interview had at least one trained facilitator and one trained scribe. However, for larger focus groups up to three additional trained scribes were also present. All scribes used a previously developed note-taking template to ensure reliability of notes across individuals. All focus group interviews were iterative, semi-structured, and allowed the facilitator to adapt questions to the participants’ perceptions and experiences.

Each focus group interview followed a predetermined question guide adapted from the recommendations of the National Institute of Diabetes and Digestive and Kidney Disorders (NDDKD) website for, “Choosing a Safe and Successful Weight-loss Program” and the Consumer Affairs website, “Best Weight Loss Programs” [38,39]. The question guide began with self-reflective questions regarding immediate changes participants would be willing to make in their lives, followed by questions about participant knowledge needs, preferred trainings to facilitate weight management, how the program should be structured, and what kinds of built-in accountability mechanisms to include (Appendix A
Table A1). The purpose of focus groups interviews was to generate a large and diverse number of responses from participants that would then be narrowed down to the most important aspects to respondents in the subsequent survey. Participants also received a brief handout consisting of exemplar weight management programs containing different program durations, number and length of meetings, modes of delivery, and meeting topic examples to help facilitate the discussion (Appendix A
Table A2 and Table A3).

### 2.4. Survey Data Collection Procedures

Following completion and analysis of the focus group interviews, a brief survey was created for member checking or participant validation purposes. Member checking provides an opportunity for participants to check or approve particular aspects of the interpretations from focus group analyses, and explore the credibility of results [40,41,42]. Thus, the survey was developed as a simple descriptive measure to help the researchers determine if the responses from data analysis were congruent with the participant experiences [42]. The survey concentrated on the most important aspects of a weight management program for study participants based on common themes identified in the focus group interviews. As such, the follow-up survey questions were similar to those used on the focus group guide to prevent misinterpretation by participants. Potential answers for each question were also taken directly from the focus group interview findings to replicate words and phrases used by participants. Questions focused on immediate changes participants would be willing to make, what type of information they would like to receive in a program, and how the program could be structured to best fit their needs. One example of a survey question about immediate change was, “If you were ready to begin a new weight management program tomorrow what is one eating habit you are most likely to change?” Potential responses to this questions included decreasing the amount of carbs, sugar, and sweets; limiting soda or sugary drinks; decreasing the amount of fast food or fried food; or establishing a routine for eating at regular times, all of which were comments that came from the focus group interviews.

Surveys were mailed to all eligible participants for completion, along with a self-addressed and stamped envelope for participants to return the completed surveys. Two weeks after administration of the survey, all participants who had not completed the mailed survey received a follow-up phone call to allow for completion of the survey by phone.

### 2.5. Analyses

Using the notes taken during each of the focus group interviews, two members of the research team completed independent qualitative content analyses to identify core consistencies across all focus group interviews. In particular, through a continuous and iterative review process for each set of notes, the two research team members identified ideas and comments that were present throughout each focus group interview to develop common codes about important topics and structural aspects of the weight management program. Once the research team members completed the individual analyses, the entire research team met to discuss the individual findings with the objective of creating a common code list. Final codes for analysis provided the following classification system: current weight management knowledge, dietary and physical activity needs, and program structure. After the common code list was determined, the two team members revisited all of the notes to reapply the codes and begin developing themes to establish the most important patient participant recommendations. The full research team then met one more time to discuss the identified themes that subsequently became the responses to include in the follow-up survey.

This series of meetings represents the triangulation process of qualitative data analysis and helped to ensure the credibility of the findings [43]. The careful and systematic completion of the individual and group level analyses and coding processes complies not only with the general qualitative methods literature, but also with the Journal Article Reporting Standards for Mixed Methods Research Design (JARS-Mixed) guidelines [43,44].

Descriptive statistics were used to analyze all survey responses using SPSS Statistics version 25. Frequency analyses were utilized to determine the most important structural components for integration into the weight management program.

## 3. Results

A total of 61 individuals participated in the focus group interviews; 52 individuals were recruited directly from the research collaborative, with another nine individuals being guests of collaborative members. The survey was sent to the original 74 research collaborative participants in addition to the nine new members who came to the focus group interviews with a friend. A total of 37 completed surveys was returned for a response rate of approximately 45%.

Table 1 presents the demographic characteristics of individuals who participated in both the focus group interviews and completed the follow-up survey. Results are reported as mean scores with standard deviation or percentages. The majority of participants were female (84%) and African-American (92%). The mean age of participants was 56.6 years (SD ± 15.3). More than half of participants (62.3%) had been diagnosed with diabetes and most participants diagnosed with diabetes had been living with the disease for ten years or less (71.1%; mean years 10.6 ± 10.1). The number of years living with diabetes ranged from 6 months to 43 years, resulting in the average number of years for the group to be 10.6 years (SD ± 10.1). Eight participants had been told they had pre-diabetes and on average had been living with pre-diabetes for 1.44 years (SD ± 0.82).

As mentioned in the methods section, responses from the focus group interviews were used to develop the follow-up survey. Thus, the remainder of this section will focus on the results gathered from the follow-up survey, which focused on participant self-reflection, components of a weight management program, and how to structure a weight management program. Questions and the corresponding participant responses from the follow-up survey are presented in Appendix A (Table A4).

For the self-reflection questions, nearly half of participants (48.6%) identified noting physical changes, such as clothes not fitting or having low energy, as key motivators for setting and completing weight loss goals. Recommendations from their doctor was also another common motivator for setting and maintaining weight loss goals among participants. The majority of participants (75.7%) felt that they would most likely be able to decrease their daily intake of both carbohydrates and sugar from food or drinks if they were beginning a weight management program. Most participants (45.9%) also identified walking or running as the most likely physical activities they would be willing to start if they began a weight management program. Attending group fitness classes or completing modified exercises at home (both 18.9%) were the second most likely physical activities for participants to start. The final self-reflection question asked patients to identify whom they would like to set program goals with, and 45.9% of participants identified that they would want help from family and/or friends.

Questions focused on participants’ diet knowledge needs, found that most respondents wanted knowledge about how to limit their carbohydrate and sugar intake (45.9%). Participants also identified how to read ingredient labels, determine portion sizes, and finding low-sugar, low-carb, and low-sodium substitutes. For knowledge about physical activity, participants were most interested in identifying different exercises that were appropriate for those with physical limitations (48.6%), which was followed by understanding what amount of physical activity is required (e.g., number of repetitions, sets, miles, days per week) (24.3%) in order to see results. Additional knowledge areas identified by participants were how to set achievable goals and remain motivated to reach those goals, along with how to effectively manage diabetes and weight during parties or around the holidays.

Moving onto questions about the structure of the weight management program, responses about the overall length of the program and frequency of meetings were somewhat evenly distributed among participants. Three months was the most common response for the length of a program with 32.4% of responses followed by twelve months (24.3%), six months (18.9%), and then twenty-four months (16.2%). Twice per month was the most commonly selected meeting frequency at 27%, with once per month and once per week following closely behind at 24.3% and 21.6%, respectively. In person meetings were the most frequently selected for meeting type (48.6%) followed by a mix of in person and phone call meetings. Most participants preferred 1 h meetings (70.3%) alternating between one-on-one and group meeting formats (67.6%). The final question on the survey asked participants about accountability throughout the program. Nearly half of respondents (48.6%) stated they would like support from a healthcare professional to help them succeed in a weight management program. The second most common answer was having a partner throughout the program to hold them accountable for reaching their goals.

## 4. Discussion

According to the responses of participants in this mixed-methods study to develop a patient-centered weight management program, the ideal program would take place over a three-month period with meetings occurring twice per month. Most participants preferred that all meetings take place in-person, with a mixture of group and individual meetings lasting approximately one hour each. The meetings should be led by a healthcare professional focusing on diet, exercise, and goal setting. Specific topics to be covered include how to limit sugar, carbohydrates, and sweets; how to find appropriate exercise routines based on individual physical limitations; and how to set goals that are achievable and remain motivated to reach those goals. The topics for each of the remaining meetings would be flexible and chosen based on specific individual or group needs within that community pharmacy population.

It is also important to note that while the above stated aspects of the ideal weight management program were preferred by most of the participants, there was not a clear majority preference for most questions. As such, three alternative community pharmacy-based weight management programs were proposed based on the findings (Table 2). Each of the programs varies in program length, and frequency of session meetings. The required number of topics for each program also varies as the number of potential meeting sessions increased based on the length of the program. The most common participant responses for diet and exercise topics are included in all potential programs. Additional topics could be added based on the next highest response from the patient survey. Any remaining session topics could be selected based on the individual or community need.

While resources and information required for successful weight management vary by individual, a level of general consistency was found between our results and previous literature. To begin, motivation was identified in our survey and previous studies, as being a potential barrier to completing a weight management program [33]. Furthermore, patients were interested in a weight management program, but, did not know how to set achievable goals and remain motivated during difficult times, like the holidays, to reach those goals [33]. Social support and accountability were also barriers identified in previous literature that were present in our results [35]. In particular, nearly half of the respondents in this survey wanted the program to involve family and/or friends in the goal setting process.

The Self-Determination Theory suggests that autonomy support when setting and achieving goals can help establish the context for the development of self-directed, personally meaningful choices and cultivate autonomous self-regulation [45]. Additionally, autonomy support, through mechanisms such as family and/or friends has been shown to facilitate weight loss progress [46]. Autonomy support can also be provided by acknowledging an individual’s perspective, refraining from excessive control and pressure, providing choices and options, and providing informational positive feedback [46].

Along with the identification of similar barriers to weight management, many of the topics identified by patients for the proposed weight management programs aligned with the curriculum of previously successful programs like the Diabetes Prevention (DPP) and PreventT2 (PT2) programs, which was created from the DPP curriculum [9,47]. For example, participants highlighted food specific topics such as reading food labels, controlling portion sizes, balancing caloric intake of sugar or carbohydrates, and staying on track during holidays or social events were areas of particular interest to patients on the survey. These topics align with the following modules from the DPP and PT2P: Three Ways to Eat Less Fat and Fewer Calories; Healthy Eating; Four Keys to Healthy Eating Out; Handling Holidays, Vacations, and Special Events; Eat Well to Prevent T2; Track Your Food; Eat Well Away from Home; More About Carbs [9,47].

Ultimately, the specific content findings both with respect to barriers faced by patients, and their knowledge needs, are not unique. However, the attention paid to the structure of a weight management program to be delivered in a community pharmacy setting and the findings gathered therein are unique. This combination of familiar and novel findings is beneficial in so far as the proposed weight management program structures can rely upon the evidence-base programs that have been previously developed, while adapting to the needs of patients [9]. Prior to implementation of any proposed program, educational materials should be compiled from current weight-management programs and training about program administration should be provided. Material from current programs is developed in a manner that allows any healthcare professional to lead the program. However, pharmacists may need additional training in nutrition and exercise topics. Collaboration with a nutritionist may also be beneficial to ensure an evidence-based approach is taken. The creation of a program that can be implemented in a community pharmacy also provides more opportunities to engage patients, especially in rural areas where other health services are often limited [16,17].

There are some limitations in this study that should be considered. To begin the study participants were drawn from a population of people participating in a previously established research collaborative for those who have diabetes, pre-diabetes, or are family members of people who have diabetes. However, given the objective of this study was to identify characteristics for the development of a future patient-centered weight management program and not to generalize the findings to the population, the impact of this limitation is minimal. Participants were also provided a handout of exemplar programs that were not previously associated with a pharmacy setting. This may have biased participant responses to align with the aspects of these previously established programs instead of truly identifying components that were important to participants or feasible in a pharmacy setting. Given the authors’ previous interactions with patient participants in this setting, providing an exemplar program was important in ensuring participants would have a foundation upon which to base their comments. Additional work is required to ensure that any program is feasible within the community pharmacy setting. Finally, the number of participants involved in this project may be considered too small. However, given the mixed methods design, and the level of detail expected through the design of both the focus group and survey questions, a larger sample size would have made identifying patterns in the data difficult. Furthermore, the mixed methods approach allowed the research team to narrow down findings from the focus group interviews with participants, providing an additional layer of validity.

Developing a weight management program that accounts for and integrates the values, needs, and desires of patients in achieving healthcare goals will be more meaningful to them, and thereafter more likely to be used [48]. By accounting for patients’ needs and wants in the design of a program, patients will be more likely to engage with all aspects of the program and ultimately more successful in weight management. The next phase of this project must be to interact with community pharmacists to determine their perspective on the structure and design of a weight management program to be delivered in their setting. Even if the needs of patients are completely met, failing to account for the needs of community pharmacists, the healthcare provider designated to deliver these services, will still result in the failure of the program.

## 5. Conclusions

Gathering patients’ perspectives on the structure and components of a weight management program is important for developing a program in which patients are willing to participate. The results from the study suggest this patient population would prefer a three-month program with a combination of group and individual in-person meetings occurring twice per month. The meetings should last no longer than one hour and be led by a healthcare professional focusing on areas of diet, exercise, and goal setting. The structure of a weight management program for delivery in a community pharmacy has not been previously examined from the patients’ perspectives. As such, these results offer a unique insight into an ideal patient-centered weight management program. As obesity continues to affect millions of individuals in the United States and worldwide, the development of weight management programs that account for patient perspectives is vital to improve engagement and ultimate patient success.

## Figures and Tables

**Table 1 pharmacy-07-00165-t001:** Participant Demographics (N = 61).

Gender	N (%)
Women	51 (83.6)
Men	10 (16.4)
**Race**	
White	6 (9.8)
African-American	55 (90.2)
**Age (years)**	
Mean + SD	56.6 ± 15.3
Range	18–88
**Years with Diabetes (n = 38)**	
Mean ± SD	10.6 ± 10.1
Range	0.5–43
**Years with Pre-Diabetes (n = 8)**	
Mean ± SD	1.44 ± 0.82
Range	0.5–3

**Table 2 pharmacy-07-00165-t002:** Examples of Patient-Informed Community Pharmacy-Based Weight Management Program Structures.

Duration of program	3 months	6 months	12 months
**Number of meetings**	12	12	18
**Type of Meeting**	6 in-person group meetings; 6 individual meetings	6 in-person group meetings; 6 individual meetings	9 in-person group meetings; 9 individual meetings
**Length of meetings**	1 h	1 h	1 h
**Frequency of meetings**	Every week	Every two weeks	Every two weeks for the first 6 months then monthly for the last 6 months
**Mode of information delivery**	Meetings with health professional; Presentation and handouts provided at meetings	Meetings with health professional; Presentation and handouts provided at meetings	Meetings with health professional; Presentation and handouts provided at meetings
**Meeting discussion topics**	**Required topics**: Introduction and baseline assessment; how to limit sweets, sugars, and carbs; understanding food labels and portion control; surviving holidays and parties; how to find low carb, sugar, and salt alternatives; appropriate exercises based on personal limitations; how to set goals and remain motivated; develop maintenance plan and goals for future **Additional discussion topics based on community needs**: Medication therapy management; calorie balance; understanding carbs, fats, and proteins; recognizing patterns and cues; overcoming barriers and preventing relapse; supermarket survival (grocery shopping)	**Required topics**: Introduction and baseline assessment; how to limit sweets, sugars, and carbs; understanding food labels and portion control; surviving holidays and parties; how to find low carb, sugar, and salt alternatives; appropriate exercises based on personal limitations; how to set goals and remain motivated; develop maintenance plan and goals for future **Additional discussion topics based on community needs**: Medication therapy management; calorie balance; understanding carbs, fats, and proteins; recognizing patterns and cues; overcoming barriers and preventing relapse; supermarket survival (grocery shopping)	**Required topics**: Introduction and baseline assessment; how to limit sweets, sugars, and carbs; understanding food labels and portion control; surviving holidays and parties; how to find low carb, sugar, and salt alternatives; appropriate exercises based on personal limitations; how to set goals and remain motivated; develop maintenance plan and goals for future. **Additional discussion topics based on community needs**: Medication therapy management; calorie balance; understanding carbs, fats, and proteins; recognizing patterns and cues; overcoming barriers and preventing relapse; supermarket survival (grocery shopping)
**Follow-up period**	24 months	24 months	24 months
**Outcomes and results tracking**	**Each meeting**: weight, success of weight loss and behavioral goals related to previous sessions. **Overall (beginning and end of program; 24-month follow-up)**: weight, BMI, waist circumference, percent body fat, percent visceral fat, and percent muscle mass, program satisfaction, change in knowledge	**Each meeting**: weight, success of weight loss and behavioral goals related to previous sessions **Overall (beginning and end of program; 24-month follow-up):** weight, BMI, waist circumference, percent body fat, percent visceral fat, and percent muscle mass, program satisfaction, change in knowledge	**Each meeting:** weight, success of weight loss and behavioral goals related to previous sessions **Overall (beginning and end of program; 24-month follow-up)**: weight, BMI, waist circumference, percent body fat, percent visceral fat, and percent muscle mass, program satisfaction, change in knowledge

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
