# Peer review of "Building a Patient-Centered Weight Management Program: A Mixed Methods Project to Obtain Patients’ Information Needs and Ideas for Program Structure"

_pharmacy, 2019, doi:10.3390/pharmacy7040165_

Round 1
Reviewer 1 Report
Thanks for the opportunity to review. I have the following suggestions:
Page 1 line 18 change to validate to validation of
page 1 line 34 change faces to face and delete killers
Page 2 line 60 need to reword sentence for flow
Page 3 line 92 ultimately will more
Page 3 line 103 Focus group interview participants
page 4 line 125 change note taker to scribe
page 7 line 245 change to to for
Reviewer 2 Report
This study focusing on the willingness of a group of pharmacy customers, who were in first line females and of African-American origin, to participate in weight loss programs and on how such programs should be structured, is written in a clear and concise manner. The English used is flawless and the text is easy to understand. Unfortunately, the mixed methods approach suffers from some intrinsic flaws, of which the low number of respondents to the survey and the application of the survey to the participants of the focus group interviews are the most relevant ones. The selection of the study participants is limited to patients diagnosed with diabetes T2 or pre-diabetes, who were pharmacy customers and mainly females of African-American origin. Therefore, the study results hardly can be generalized to the general population. An Appendix with the text of the applied questionnaire is missing.
Comments in detail:
Title: All findings in this manuscript are limited to individuals who expressed their interest in pharmacy-based weight management programs. Moreover, most (if not all) of the focus group interview participants were patients from an existing research collaborative focused on diabetes self-management and were visiting a pharmacy. Almost all of the participants were women of African-American ethnicity, limiting the generalization of the study further. These constraints, as listed in l. 103-110, should be considered in the Title, which should be more specific indicating these limitations.
l. 34 Rephrase ”chronic disease killers” – might be misunderstood as killers of chronic disease”.
l. 68-72 The arguments that pharmacists have a high potential to access large parts of the population and that parts of the population would be willing to accept being guided through weight-loss programs by pharmacists has to be accepted. However, the adequacy of pharmacists’ education in guiding patients suffering from obesity has to be taken into doubt. Dietary advice is traditionally not part of the curriculum during the education of pharmacist and professionals specifically educated in the treatment and prevention of obesity have to be considered much better qualified. From the manuscript, it does not become clear to which extent and how the weight loss program should be communicated to customers visiting a pharmacy. Do the authors assume that a health professional qualified to treat obese patients should be present all the time in the pharmacy? The underlying claim that pharmacists would be the right professional group to treat obesity should be removed from the manuscript.
l. 129-139 Seemingly, the focus group interviews were hedged by guidelines and directives extracted from other running programs on obesity management that were not or are not associated with pharmacies. Nevertheless, the focus group interviews were managed in front of a pharmacy background and all answers given by the study participants have to be considered as biased because of this setting.
l. 148-152 It remains unclear on which scientific evidence basis the authors selected the phrasing of the question, as briefly described in l. 148-149. How do the authors know that these changes would result in a scientifically valid approach to reduce weight?
l. 141-156 This section lacks relevant information on the number of questions included, whether attitudes were measured by using Likert scales, and if so, how many levels were used for those. It’s unclear whether multiple responses were allowed. Also, it remains unclear whether measurements of attitudes were split up into item statements and whether Cronbach’s alpha was calculated for those. No power estimation of the quantitative study part has been reported. A brief indication of degrees of freedom is needed in face of the low number of survey respondents. In general, the number of respondents to the survey is too low to provide reliable results. Acknowledging this limitation (l. 304-305) does not help to overcome this flaw.
l. 142-144 A survey should reflect the unbiased, original opinion of the target group in question. Participation in focus groups is coining the opinions of the participants, making their mindset quite different from people who did not participate in such interviews. Therefore, the survey should not have included the participants of the focus group interviews.
l. 183 It’s uncommon to use percentages when counts are below 100.
l. 226-229 Although the authors did not provide the exact wording of the questionnaire, it seems to me that some of the points addressed were quite unspecific. For example, how was the term “results” (l. 229) defined?
l. 308-309 As the opinion of the study participants might have been coined by the focus group discussions already, I don’t think that one can say that the application of the survey has “verified” the findings of the focus group interview.
In the Discussion, I miss a suggested strategy on how the entire model of health professional-pharmacist interaction should work. What should be the motivation of the pharmacists to invest time and effort into facilitating such weight loss programs?
Reviewer 3 Report
1) Title: make it more informative, include study design.
2) Check if abstact should be structured or not: generally, published articles in Pharmacy (Basel) have an unstructured abstract.
3) Line 42, add a comma between population and developing.
4) Line 51: check if remains elusive should be remain elusive.
5) Introduction is fine and exhaustive - may be a little bit longer. (consider to move some portions to the discussion section) I would some sentences about the importance of patient-reported outcomes in the research field, before the aims (final section of discussion).
6) Authors used the ENTREQ, but since the design is a mixed methods study, I strongly recommend authors to use designs the Mixed Methods Design Reporting Standards (JARS-Mixed) guidelines.
7) I would expand and explain better the process of coding.
8) I would expand results section, which, as it is now, results too frugal.
Round 2
Reviewer 2 Report
In the current version of their manuscript, the authors have addressed most of the points raised. However, the most relevant methodological flaws of the study persist: given the number of questions addressed, the number of respondents is too low and, instead of applying the questionnaires to an unbiased sample, the participants of the focus group interviews were re-addressed. Therefore, finally it is up to the Editorial Office to decide whether this scientific standard meets their criteria as, by format and presentation, I consider the manuscript has exploited its potential for improvement. I appreciated that these points are now addressed in the Limitations of the study section but this, naturally, does not overcome them.
I still do have some ethical concerns as the paper reminds me more of a strategy paper on how pharmacists can acquaint new customer groups than observational research but maybe in science there should also be space for expressing such an opinion. In any case, I miss in the paper a statement that pharmacists may need additional training in nutrition or should collaborate with nutritionists to provide an evidence-based approach. Please add this.
However, the manuscript is now in a formally appropriate shape to be published in a scientific peer-reviewed journal.
Minor: In Table 1, the percentage of men should say 16.4 and not 15.4.
Reviewer 3 Report
No further comments. Manuscript has been significantly improved.
